# A Novel Si Nanosheet Channel Release Process for the Fabrication of Gate-All-Around Transistors and Its Mechanism Investigation

**DOI:** 10.3390/nano13030504

**Published:** 2023-01-27

**Authors:** Xin Sun, Dawei Wang, Lewen Qian, Tao Liu, Jingwen Yang, Kun Chen, Luyu Wang, Ziqiang Huang, Min Xu, Chen Wang, Chunlei Wu, Saisheng Xu, David Wei Zhang

**Affiliations:** 1State Key Laboratory of ASIC and System, School of Microelectronics, Fudan University, Shanghai 200433, China; 2Shanghai Integrated Circuit Manufacturing Innovation Center Co., Ltd., Shanghai 201202, China; 3Zhangjiang Fudan International Innovation Center, Shanghai 200433, China

**Keywords:** nanosheet, gate-all-around, transistor, channel release, isotropic dry etch, high selectivity, deformation, plasma-free

## Abstract

The effect of the source/drain compressive stress on the mechanical stability of stacked Si nanosheets (NS) during the process of channel release has been investigated. The stress of the nanosheets in the stacking direction increased first and then decreased during the process of channel release by technology computer-aided design (TCAD) simulation. The finite element simulation showed that the stress caused serious deformation of the nanosheets, which was also confirmed by the experiment. This study proposed a novel channel release process that utilized multi-step etching to remove the sacrificial SiGe layers instead of conventional single-step etching. By gradually releasing the stress of the SiGe layer on the nanosheets, the stress difference in the stacking direction before and after the last step of etching was significantly reduced, thus achieving equally spaced stacked nanosheets. In addition, the plasma-free oxidation treatment was introduced in the multi-step etching process to realize an outstanding selectivity of 168:1 for Si_0.7_Ge_0.3_ versus Si. The proposed novel process could realize the channel release of nanosheets with a multi-width from 30 nm to 80 nm with little Si loss, unlocking the full potential of gate-all-around (GAA) technology for digital, analog, and radio-frequency (RF) circuit applications.

## 1. Introduction

As the technology node scales down to 3 nm nodes and beyond, the gate-all-around field effect transistor (GAA-FET) made of vertically stacked horizontal nanosheets is considered the most promising candidate for FinFET replacement [1,2,3,4]. First of all, GAA-FET has superior electrostatic controllability and improved short-channel effects suppression [5]. In addition, the design flexibility of GAA-FET allows for a continuous range of nanosheet widths to satisfy different requirements in a single chip with a small areal cost [6]. GAA-FET can be fabricated with minimal changes from FinFET integration [7]. The dominant scheme for fabricating vertically stacked Si nanosheets includes the epitaxial growth of the Si/SiGe multilayer structure on the Si substrate and the channel release process that involves the selective removal of the sacrificial SiGe layers. In order to achieve the co-optimization of performance and power, the simultaneous channel release of nanosheets with a multi-width is required, which requires the channel release process to have a high etching selectivity of SiGe over Si.

When the transistor architecture changes from FinFET to GAA-FET, the main conduction surface orientation is changed from the crystallographic plane (110) to (100), which results in an increase in the electron mobility but a decrease in the hole mobility [8]. To achieve N/P current matching in GAA-FET, it is necessary to introduce compressive stress in the p-channel metal–oxide–semiconductor (PMOS) channel to enhance the hole mobility [9,10,11]. The selective epitaxial growth of SiGe in the source and drain is recognized as one of the most effective methods for applying uniaxial compressive stress into the PMOS channel [12,13]. According to the mainstream GAA-FET fabrication process flow, the process of channel release is performed after the selective epitaxial growth of the SiGe source and drain [1]. Therefore, the Si nanosheets are subjected to compressive stress from the source and drain during the process of channel release. It is reported that Young’s modulus of Si nanosheets is much smaller than that of bulk Si [14,15], which means that Si nanosheets are more easily deformed under stress. Logic devices with shorter channel lengths may be less affected by sustained compressive stress during the process of channel release. In addition to logic devices, there are analog, RF, and input/output (I/O) devices on the same piece, all of which have channel lengths of more than 100 nm [16,17]. In published papers, as far as we know, there are few studies about the influence of the source/drain compressive stress on the mechanical stability of nanosheets during the channel release process.

According to previous studies, available processes for the isotropic selective etching of SiGe include wet etching [18], plasma dry etching developed in reactive ion etching [19,20] and remote plasma dry etching [21]. Wet etching is unstable in a high-aspect-ratio structure due to the influence of the capillary effect [22]. Another issue is that the capillary forces inherent in wet etching may cause adjacent nanosheets to stick together [23]. The plasma dry etching process developed in reactive ion etching will cause plasma damage to the surface of Si nanosheets, which will seriously degrade the electrical performance of the device. Remote plasma dry etching is a pure chemical etching process without plasma damage. In this work, remote plasma dry etching was used as the isotropic etching to remove the SiGe layers.

In this paper, we investigated the effect of compressive stress on the mechanical stability of stacked Si nanosheets during the channel release process. The Sentaurus simulation indicates that the stress in the direction of stacking increases first and then decreases during the channel release. The finite element simulation and experimental results show that the compressive stress applied to the channel can cause serious deformation of the suspended Si nanosheets in the direction of stacking. Subsequently, a novel channel release process was proposed which utilizes multi-step etching instead of the single-step process to remove the sacrificial SiGe layers. The novel channel release process could effectively solve the serious deformation problem by gradually releasing the stress on the nanosheets. Furthermore, oxidation treatment was introduced in the optimized channel release process to further improve the etching selectivity of SiGe over Si. The proposed novel process could realize the channel release of nanosheets with multiple widths from 30 nm to 80 nm with little Si loss.

## 2. Materials and Methods

The main fabrication process flow of vertically stacked horizontal nanosheets is shown in Figure 1a. First, three cycles of the Si/SiGe multilayer were deposited on an eight-inch Si substrate using a reduced pressure chemical vapor deposition (RPCVD) apparatus, where the thickness of the Si and SiGe layers was around 9 nm, and the Ge composition in the SiGe layers was 30% (the Si/SiGe multilayer wafer used in this paper was a commercially available wafer). Then, a hard mask was grown on the Si/SiGe multilayer, and the designed test pattern was transferred to the hard mask. The designed test pattern was transferred to the Si/SiGe multilayer structure by anisotropic etching using the inductively coupled plasma (ICP) machine. Next, the hard mask was removed by wet etching using a 1% hydrofluoric acid solution, while the native oxide was removed. After that, the sacrificial SiGe layers of the channel were removed by isotropic etching, using the remote plasma dry etching apparatus (NAURA HSE200C) to form the suspended vertically stacked Si nanosheets. Finally, the high K dielectric and metal gate (HKMG) were uniformly wrapped on the nanosheets by atomic layer deposition (ALD) technology. The cross-section of the channel was analyzed by a high-resolution transmission electron microscope (TEM) to check the mechanical stability of the stacked nanosheets. As shown in Figure 1b, the designed test pattern consists of two large pads and a channel. The parameters of the test pattern are summarized in Figure 1c, where different values of the width of the nanosheet are designed to meet the co-optimization of high-computing-performance and low-power-consumption transistors on a single chip. The large pads of the test pattern can act as stressors to introduce compressive stress into the channel. In this way, the selective epitaxial SiGe source/drain process can be omitted when studying the effect of compressive stress on the mechanical stability of the stacked nanosheets.

The Sentaurus TCAD simulator was used to obtain the evolution of the stress of the Si layers in the process of fabricating the nanosheets. Figure 2a shows the calculated stress value of the Si layer in the middle of the Si/SiGe multilayer structure along the channel direction. In the supperlattice wafer stage, no stress exists in the Si layers, and a compressive stress of about 2 GPa exists in the SiGe layers. After the test pattern is transferred to the multi-layer structure, the emergence of free edges in the structure results in stress relaxation [24]. Then, tensile stress is generated in the Si layers of the pad, and a compressive stress of about 0.3 GPa is generated in the Si layers of the channel. The tensile stress in the Si layers of the pad is caused by the partial relaxation of the compressive stress in the SiGe layers [25,26]. The compressive stress in the Si layers of the channel is the result of the lattice in the horizontal direction becoming larger when the Si layers of the pad are subjected to tensile stress. After the channel release, the compressive stress of the Si layers of the channel increases to about 0.8 GPa. As shown in Figure 2b, it can be seen intuitively that there is tensile stress in the Si layers of the Pad, and there is compressive stress in the SiGe layers of the Pad and the Si layers of the channel.

## 3. Results and Discussion

The stress values of the nanosheets in the stacking direction were extracted by the Sentaurus simulator. In the data extraction, the resultant force of the stresses on the upper and lower surfaces of each nanosheet in the stacking direction was considered as the stress on the corresponding nanosheet. Figure 3a shows the evolution of the stress on the nanosheets with a width of 60 nm as the etching amount of the SiGe layers increases. During the channel release process, the direction of stress on each nanosheet in the stacking direction was unchanged. The direction of stress on the top nanosheet was downward, and the direction of stress on the middle and bottom nanosheets was upward. With the increase in the SiGe etching amount, the stress of the nanosheets first increases and then decreases. The maximum stresses on the three nanosheets from top to bottom were 1.42 MPa, 3.86 MPa, and 9.1 MPa, respectively. When the SiGe layers of the channel were completely removed, the stress tended to a constant value, which was too small to cause deformation of the nanosheets. The traditional channel release process completely removes the SiGe layers of the channel by single-step etching. The process has a short execution time, so it can be considered a transient process. For the traditional channel release process, the maximum stress extracted during the process can be regarded as the stress in the stacking direction of the nanosheet when the sacrificial layer is completely removed. The deformation of stacked nanosheets caused by the maximum stress values obtained by the finite element simulation model is shown in the inset of Figure 3a. Figure 3b shows the TEM cross-sectional image of stacked nanosheets with a width of 60 nm obtained by single-step etching, which is consistent with the simulation. The mechanical structure of the device was damaged, in which two adjacent nanosheets were squeezed together. The unequal spacing between nanosheets results in differences in the thickness of the subsequently deposited work function metal, which results in differences in device threshold voltages [27]. To obtain GAA-FETs with a high electrical performance, it is expected to obtain stacked nanosheets with uniform spacing between adjacent nanosheets.

The channel release process was changed from the traditional single-step etching to multi-step etching, which was equivalent to performing multiple etching to remove the SiGe layers of the channel. In this way, when the nanosheets were subjected to stress in the stacking direction that could cause severe deformation during the multi-step etching process, the remaining sacrificed SiGe layers played a supporting role to prevent the nanosheets from deformation. By gradually releasing the stress of the SiGe layer on the nanosheets, the stress difference in the stacking direction before and after the last step of etching was reduced. Therefore, when the SiGe layers were completely removed after the last etching step, the stress on the nanosheets during the process was not enough to cause significant deformation of the nanosheets. In practice, we applied this concept of multi-step etching in releasing 60 nm-wide nanosheets. As shown in Figure 3c, the stacked nanosheets obtained by multi-step etching are almost equally spaced. The result shows that multi-step etching can effectively solve the problem of nanosheets deformation caused by the source/drain compressive stress during the channel release.

When Si and SiGe co-exist at the beginning of remote plasma dry etching, fluorine neutral species preferentially react with SiGe due to the higher Si-Si bond energy (2.31 eV) compared to the Si-Ge bond energy (2.12 eV) [28], achieving a high selective removal of SiGe layers. Once the SiGe has been completely removed, the Si etching rate increases dramatically [28]. This would cause a serious problem of Si loss when there are multiple widths of nanosheets on the single chip in practice. As can be seen from the TEM images of stacked nanosheets shown in Figure 3b,c, the cross-section nanosheets exhibit a fusiform shape due to the obvious Si loss at the entrance of nanosheets caused by over-etching.

Based on the concept of multi-step etching, oxidation treatment was introduced to solve the problem of the serious Si loss caused by inevitable over-etching. After the oxidation treatment, an oxide layer will be formed on the surface of Si and SiGe, which has an inhibitory effect on the etching reaction. The previous study has demonstrated that the removal time of the oxide layer on the Si surface is much longer than that on the SiGe surface in the etching process based on fluorine-neutral species [29]. Different from the oxygen plasma oxidation process, the process adopted in this paper was a method similar to natural oxidation, which can completely avoid plasma damage. The specific operation is to directly feed a mixture of oxygen and argon into the reaction chamber, wherein the remote plasma source is turned off. It was demonstrated that the introduction of oxidation treatment in multi-step etching could achieve a very high etching selectivity of SiGe to Si by experiment. A selectivity of ~168:1 for our novel channel release process is determined using long etch times on Si_0.7_Ge_0.3_ for an extremely wide nanosheet. Selectivity is defined as the ratio between the SiGe tunnel depth (*d*) and the Si layer thickness loss ((*a* − *b*)/2) once the tunnel configuration is achieved, that is, $Selectivity=\frac{d}{\left(a-b\right)/2}$, which is illustrated in Figure 4. As shown in Figure 4, the SiGe is laterally consumed of 41.93 nm, whereas the etching Si nanosheet sidewall is consumed of 0.25 nm.

Since the oxide layer on the Si surface is generated by consuming Si, the number of oxidation treatments should be reduced as much as possible to avoid excessive Si consumption at the entrance of the nanosheets. According to the characteristics of the fluorine-based remote plasma etching process, for the channel release of nanosheets with a single width, as shown in Figure 5, the oxidation treatment is only necessarily performed before the last etching step of the multi-step etching process. For the simultaneous channel release of nanosheets with multiple widths, oxidation treatment would be needed before the last etching step of the nanosheet of each width. Figure 6 shows the TEM cross-sectional images of stacked nanosheets with multiple widths obtained by the multi-step etching with oxidation treatment. The Si loss at the entrance of the nanosheets is significantly suppressed, and the cross-section of the nanosheets tends to be more square-like.

## 4. Conclusions

In this paper, the effect of the source/drain compressive stress on the mechanical stability of stacked Si nanosheets during the process of channel release is investigated. For the traditional channel release process, both the simulation and experiments show that the compressive stress applied to the channel can cause serious deformation of the suspended Si nanosheets in the stacking direction. A novel channel release process utilizing multi-step etching to remove the sacrificial SiGe layers was proposed in this work. By gradually releasing the stress of the SiGe layer on the nanosheets, equally spaced stacked nanosheets were successfully achieved. The related mechanism was explained by the TCAD simulation and validated in the experiment. Additionally, a plasma-free oxidation treatment was introduced in the multi-step etching process to realize an outstanding selectivity of 168:1 for Si_0.7_Ge_0.3_ over Si. With the channel integrity and extraordinary etch selectivity, the proposed process realized the channel release of nanosheets with a multi-width from 30 nm to 80 nm, which showed great potential in the GAA technology.

## Figures and Tables

**Figure 1 nanomaterials-13-00504-f001:**
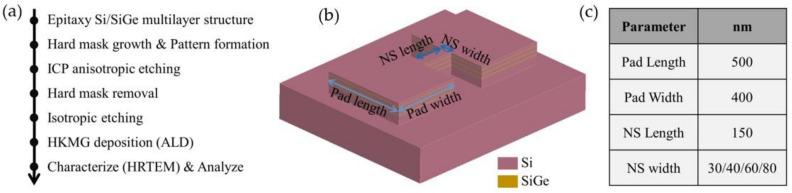
(**a**) The main fabrication process flow of vertically stacked horizontal nanosheets. (**b**) 3D schematic diagram of the test pattern after the step of hard mask removal. (**c**) Summary of the parameters of the test pattern. (Inductively coupled plasma: ICP; High k dielectric and metal gate: HKMG; Atomic layer deposition: ALD).

**Figure 2 nanomaterials-13-00504-f002:**
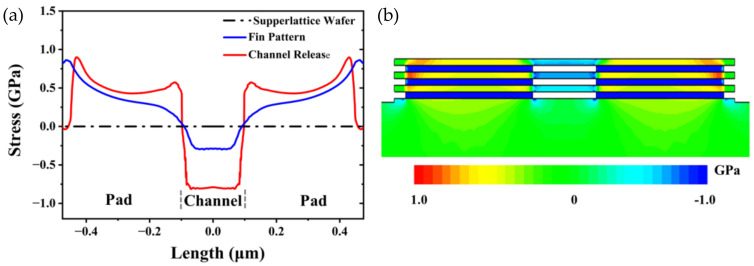
(**a**) Calculated stress values along the Si channel direction in the middle of the Si/SiGe multilayer structure obtained by the Sentaurus simulation. (**b**) Stress distribution in the test pattern after the channel release obtained by the Sentaurus simulation.

**Figure 3 nanomaterials-13-00504-f003:**
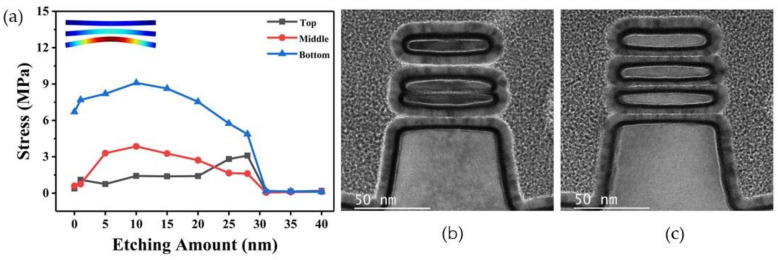
(**a**) The stress evolution on the 60 nm-wide nanosheets along with the increasing etching amount of the SiGe layers. Inset: The deformation of the nanosheets caused by the maximum stress values obtained by finite element simulation. TEM cross-sectional images of 60 nm-wide stacked nanosheets obtained by (**b**) single-step etching and (**c**) multi-step etching.

**Figure 4 nanomaterials-13-00504-f004:**
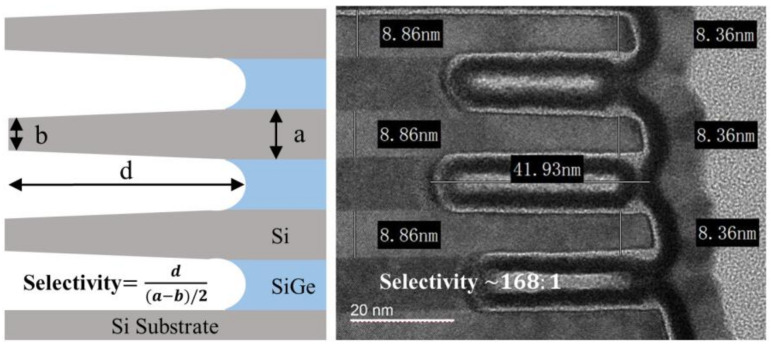
SiGe versus Si etching selectivity of multi-step etching with the application of oxidation treatment.

**Figure 5 nanomaterials-13-00504-f005:**
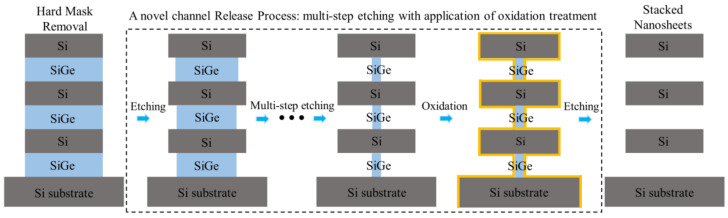
Schematic diagram of the multi-step etching process for the channel release of single-width nanosheets with the application of oxidation treatment.

**Figure 6 nanomaterials-13-00504-f006:**
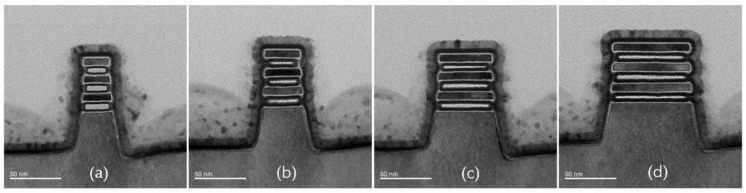
TEM cross-sectional images of stacked nanosheets with multiple widths obtained by multi-step etching with the application of oxidation treatment. (**a**) 30 nm; (**b**) 40 nm; (**c**) 60 nm; (**d**) 80 nm.

## Data Availability

The data presented in this study are available on request from the corresponding authors.

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
