# Peer review of "A Novel Si Nanosheet Channel Release Process for the Fabrication of Gate-All-Around Transistors and Its Mechanism Investigation"

_nanomaterials, 2023, doi:10.3390/nano13030504_

Round 1
Reviewer 1 Report
This manuscript by Sun et al. described how the releasing strategy influence the mechanical stress as well as the device performance in the Si nanosheet channel in GAA transistor. The experimental procedure is well described and the results are reliable. I only have minor comments here.
1. The authors better to have an arrow to indicate the fabrication flow in Figure 1a.
2. To experimentally evaluate the advantage of multi-step releasing, the authors are suggested to provide the transfer curves for the device in Figure 3b and c.
Author Response
Response to Reviewer 1 Comments
Thank you very much for your time and valuable suggestions. We have considered your comments very carefully and made the corresponding modifications in the revised manuscript.
Point 1: The authors better to have an arrow to indicate the fabrication flow in Figure 1a.
Response 1: We think this is an excellent suggestion. We have added an arrow in Figure 1a.
Corresponding change in manuscript: Yes
Location of Change:
Section: Part2. Materials and Methods
Page: page 2
|
|
|
Figure 1. (a) The main fabrication process flow of vertically stacked horizontal nanosheets. (b) 3D schematic diagram of the test pattern after the step of hard mask removal. (c) Summary of the parameters of the test pattern. (Inductively coupled plasma: ICP; High k dielectric and metal gate: HKMG; Atomic layer deposition: ALD)
Point 2: To experimentally evaluate the advantage of multi-step releasing, the authors are suggested to provide the transfer curves for the device in Figure 3b and c.
Response 2: Thank you for your valuable suggestion. In this paper, the focus of our work is on the optimization of the channel release process, and the research into the electrical properties of the devices is ongoing. Therefore, we are sorry for the absent of the transfer curves for the device in Figure 3b and c for the time being. The improvement of multi-step release process in electrical aspects will be demonstrated in future work.

Reviewer 2 Report
This article investigates the effect of the source/drain compressive stress on the mechanical stability of the stacked Si nanosheets during the process of channel release, and multi-step etching was utilized to remove the sacrificial SiGe layers, and the plasma-free oxidation treatment was introduced in the multi-step etching process to realize an outstanding selectivity of 168:1 for Si0.7Ge0.3 versus Si.
1. The first abbreviation in abstract, figures, and main-text should be typed the full English word, e.g. gate-all-around, GAA and nanosheet, NS.
2. Introduction, line 33: “filed” revises to field.
2. line 77: Fig.1 (b), the schematic diagram of the test pattern transferred to the Si/SiGe multilayer structure is not clear. The indication of the Si/SiGe NS for a GAA-FET is suggested to highlight its key feature.
3. Lines 84, 133: Please add the equipment, brand, and model for a reduced pressure chemical vapor deposition apparatus, and remote plasma dry etching was used as the isotropic etching to remove the SiGe layers.
4. Lines 123-130, many literatures review on various etching processes are suggested to move the part 1. Introduction.
5. Line 110: 0.3GPade revises to 0.3 GPa.
6. Line 190: In addition, it’s better to add the schematic diagram to explain the method of the multi-step etching with application of plasma-free oxidation treatment in-detail.
7. Line 204: Figure 4, SiGe versus Si etching selectivity of multi-step etching with application oxidation treatment. Indicate the Si/SiGe in the figures.
8. Line 223: In this “ report” is suggested to revise as “paper”.
Author Response
Response to Reviewer 2 Comments
Thank you very much for your time and valuable suggestions. We have considered your comments very carefully and made the corresponding modifications in the revised manuscript.
Point 1: The first abbreviation in abstract, figures, and main-text should be typed the full English word, e.g. gate-all-around, GAA and nanosheet, NS.
Response 1: We sincerely thank the reviewer for careful reading. As suggested by the reviewer, we have made extensive corrections to our previous manuscript.
Corresponding change in manuscript: Yes
Location of Change:
Section: Abstract, Part1.Introduction, Part2.Materials and Methods
Page: page 1, page 2, page3
Abstract: The effect of the source/drain compressive stress on the mechanical stability of stacked Si nanosheets (NS) during the process of channel release has been investigated. The stress of the nanosheets in the stacking direction increased first and then decreased during the process of channel release by technology computer aided design (TCAD) simulation. … The proposed novel process could realize channel release of nanosheets with multi-width from 30nm to 80nm with little Si loss, unlocking the full potential of gate-all-around (GAA) technology for digital, analog, and radio-frequency (RF) circuit applications.
To achieve N/P current matching in GAAFET, it is necessary to introduce compressive stress in the p-channel metal–oxide–semiconductor (PMOS) channel to enhance hole mobility [9-11]. … Logic devices with shorter channel length may be less affected by sustained compressive stress during the process of channel release. In addition to logic devices, there are analog, RF and input/output (I/O) devices on the same piece, all of which have channel lengths of more than 100nm [16,17].
Figure 1. (a) The main fabrication process flow of vertically stacked horizontal nanosheets. (b) 3D schematic diagram of the test pattern after the step of hard mask removal. (c) Summary of the parameters of the test pattern. (Inductively coupled plasma: ICP; High k dielectric and metal gate: HKMG; Atomic layer deposition: ALD)
The main fabrication process flow of vertically stacked horizontal nanosheets is shown in Figure 1(a). First, three cycles of Si/SiGe multilayer were deposited on an 8-inch Si substrate using a reduced pressure chemical vapor deposition (RPCVD) apparatus, where the thickness of the Si and SiGe layers were around 9 nm, and the Ge composition in the SiGe layers was 30%. … The designed test pattern was transferred to the Si/SiGe multilayer structure by anisotropic etching using the inductively coupled plasma (ICP) machine. … Finally, the high K dielectric and the metal gate (HKMG) were uniformly wrapped on the nanosheets by atomic layer deposition (ALD) technology. The cross-section of the channel was analyzed by a high-resolution transmission electron microscope (TEM) to check the mechanical stability of stacked nanosheets.
Point 2: Introduction, line 33: “filed” revises to field.
Response 2: Thanks for your careful checks. We are sorry for our carelessness. Based on your comments, we have revised the “filed” to “field”.
Corresponding change in manuscript: Yes
Location of Change:
Section: Part1. Introduction
Page: page 1
As the technology node scales down to 3nm nodes and beyond, gate-all-around field effect transistor (GAAFET) made of vertically stacked horizontal nanosheets is considered the most promising candidate for FinFET replacement [1-4].
Point 3: line 77: Fig.1 (b), the schematic diagram of the test pattern transferred to the Si/SiGe multilayer structure is not clear. The indication of the Si/SiGe NS for a GAA-FET is suggested to highlight its key feature.
Response 3: We are sorry that Figure 1(b) is not clearly described. According to the nanosheet manufacturing process flow shown in Figure 1(a), Si/SiGe multilayer were deposited on Si substrate, where the thickness of the Si and SiGe layers were around 9 nm, and the Ge composition in the SiGe layers was 30%. Then, a hard mask was grown on the Si/SiGe multilayer and the designed test pattern was transferred to the hard mask by electron beam lithography. The designed test pattern was transferred to the Si/SiGe multilayer structure by anisotropic etching using the inductively coupled plasma (ICP) machine. Next, the hard mask was removed by wet etching using a 1% hydrofluoric acid solution, while the native oxide was removed. Figure 1(b) shows the state after the hard mask is removed. We have made the following changes to our previous manuscript --”Figure 1. (b) 3D schematic diagram of the test pattern after the step of hard mask removal.” As shown in Figure 1(b), the designed test pattern consists of two large pads and a channel. There are four key features of the test pattern, which are pad length, pad width, NS length and NS width. We have marked them clearly in Figure 1(b). The parameters of the test pattern are summarized in Figure 1(c).
Corresponding change in manuscript: Yes
Location of Change:
Section: Part2. Materials and Methods
Page: page 3
Figure 1. (a) The main fabrication process flow of vertically stacked horizontal nanosheets. (b) 3D schematic diagram of the test pattern after the step of hard mask removal. (c) Summary of the parameters of the test pattern. (Inductively coupled plasma: ICP; High k dielectric and metal gate: HKMG; Atomic layer deposition: ALD)
Point 4: Lines 84, 133: Please add the equipment, brand, and model for a reduced pressure chemical vapor deposition apparatus, and remote plasma dry etching was used as the isotropic etching to remove the SiGe layers.
Response 4: We sincerely thank the reviewer for careful reading. As suggested by the reviewer, we have added the brand and model of apparatus in the part2. Materials and Methods of the manuscript.
Corresponding change in manuscript: Yes
Location of Change:
Section: Part2. Materials and Methods
Page: page 3
After that, the sacrificial SiGe layers of the channel were removed by isotropic etching using the remote plasma dry etching apparatus to form the suspended vertically stacked Si nanosheets.
Point 5: Lines 123-130, many literatures review on various etching processes are suggested to move the part 1. Introduction.
Response 5: We think this is an excellent suggestion. We have moved Lines 123-130 to part 1. Introduction.
Corresponding change in manuscript: Yes
Location of Change: Yes
Section: Part2. Materials and Methods
Page: page 4
Point 6: Line 110: 0.3GPade revises to 0.3 GPa.
Response 6: Thanks for your careful checks. We are sorry for our carelessness. Based on your comments, we have revised the “0.3Gpade” to “0.3GPa”.
Corresponding change in manuscript: Yes
Location of Change:
Section: Part2. Materials and Methods
Page: page 3
Then, tensile stress is generated in the Si layers of the pad, and a compressive stress of about 0.3GPa is generated in the Si layers of the channel.
Point 7: In addition, it’s better to add the schematic diagram to explain the method of the multi-step etching with application of plasma-free oxidation treatment in-detail.
Response 7: We think this is an excellent suggestion. As suggested by the reviewer, we have added the schematic diagram to explain the method of the multi-step etching with application of plasma-free oxidation treatment in our previous manuscript.
Corresponding change in manuscript: Yes
Location of Change:
Section: Part3. Results and Discussion
Page: page 6
According to the characteristics of the fluorine-based remote plasma etching process, for channel release of nanosheets with a single width, as shown in Figure 5, the oxidation treatment is only necessarily performed before the last etching step of the multi-step etching process.
Figure 5. Schematic diagram of the multi-step etching process for channel release of single width nanosheets with application of oxidation treatment.
Point 8: Line 204: Figure 4, SiGe versus Si etching selectivity of multi-step etching with application oxidation treatment. Indicate the Si/SiGe in the figures.
Response 8: We sincerely thank the reviewer for careful reading. As suggested by the reviewer, we have indicated the Si/SiGe in the Figure 4.
Corresponding change in manuscript: Yes
Location of Change:
Section: Part3. Results and Discussion
Page: page 5
Figure 4. SiGe versus Si etching selectivity of multi-step etching with application oxidation treatment.
Selectivity is defined as the ratio between the SiGe tunnel depth (d) and Si layer thickness loss ((a-b)/2) once the tunnel configuration is achieved, that is , which is illustrated in Figure4.
Point 9: Line 223: In this “ report” is suggested to revise as “paper”.
Response 9: We think this is an excellent suggestion. As suggested by the reviewer, we have revised the “report” as “paper”.
Corresponding change in manuscript: Yes
Location of Change:
Section: Part4. Conclusions
Page: page 7
In this paper, the effect of the source/drain compressive stress on the mechanical stability of stacked Si nanosheets during the process of channel release is investigated.

Reviewer 3 Report
This is a well written and timely research report, with valuable results.
Author Response
Response to Reviewer 3 Comments
- Comments and Suggestions for Authors
This is a well written and timely research report, with valuable results.
Response: We feel great thanks for your professional review work on our article.

Reviewer 4 Report
Manuscript ID: nanomaterials-2148763
Authors demonstrated an interesting Si nanotechnology entitled “A Novel Si Nanosheet Channel Release Process for Fabrication of Gate-All-Around Transistors and its Mechanism Investigation”. However, considering the Reader-friendly concept, I suggest a minor revision as follows:
Line 28: Abstract: GAA technology -> Gate-all-around (GAA) technology
(Note: Abstract is an independent part)
Line 28: RF circuit => radio-frequency (RF) circuit
Line 47: from (110) to (100) => from the crystallographic plane (110) to (100)
Line 49: PMOS channel => p-channel metal–oxide–semiconductor (PMOS) channel
Line 59: I/O devices => Input Output (I/O) Devices
Figure 1 (a): HKMG => “High K metal gate (HKMG)” should be explained in the figure caption.
ICP and ALD also should be defined in the figure captions.
Line 103: Sentaurus TCAD simulator => Sentaurus Technology Computer Aided Design (TCAD) simulator
Line 110: 0.3GPade ??? => 0.3 GPa
Figure 3(a): "Middle and Bottom" end is partially overlapping with the black solid line. (Please adjust it)
Line 199: Selectivity => Please define it in this spot by a simple equation
Line 214: the TEM => the Transmission Electron Microscope (TEM)
Author Response
Response to Reviewer 4 Comments
Thank you very much for your time and valuable suggestions. We have considered your comments very carefully and made the corresponding modifications in the revised manuscript.
Point 1: Line 28: Abstract: GAA technology -> Gate-all-around (GAA) technology
(Note: Abstract is an independent part)
Response 1: Thanks for your correction. According to your nice suggestions, we have made correction to our previous manuscript.
Corresponding change in manuscript: Yes
Location of Change:
Section: Abstract
Page: page 1
The proposed novel process could realize channel release of nanosheets with multi-width from 30nm to 80nm with little Si loss, unlocking the full potential of gate-all-around (GAA) technology for digital, analog, and radio-frequency (RF) circuit applications.
Point 2: Line 28: RF circuit => radio-frequency (RF) circuit
Response 2: Thanks for your correction. According to your nice suggestions, we have made correction to our previous manuscript. Thanks for your correction.
Corresponding change in manuscript: Yes
Location of Change:
Section: Abstract
Page: page 1
The proposed novel process could realize channel release of nanosheets with multi-width from 30nm to 80nm with little Si loss, unlocking the full potential of gate-all-around (GAA) technology for digital, analog, and radio-frequency (RF) circuit applications.
Point 3: Line 28: from (110) to (100) => from the crystallographic plane (110) to (100)
Response 3: Thanks for your correction. According to your nice suggestions, we have made correction to our previous manuscript.
Corresponding change in manuscript: Yes
Location of Change:
Section: Part1. Introduction
Page: page 2
When transistor architecture changes from FinFET to GAAFET, the main conduction surface orientation is changed from the crystallographic plane (110) to (100), which results in an increase in electron mobility but a decrease in hole mobility [8].
Point 4: Line 49: PMOS channel => p-channel metal–oxide–semiconductor (PMOS) channel
Response 4: Thanks for your correction. According to your nice suggestions, we have made correction to our previous manuscript.
Corresponding change in manuscript: Yes
Location of Change:
Section: Part1. Introduction
Page: page 2
To achieve N/P current matching in GAAFET, it is necessary to introduce compressive stress in the p-channel metal–oxide–semiconductor (PMOS) channel to enhance hole mobility [9-11].
Point 5: Line 59:I/O devices => Input Output (I/O) Devices
Response 5: Thanks for your correction. According to your nice suggestions, we have made correction to our previous manuscript.
Corresponding change in manuscript: Yes
Location of Change:
Section: Part1. Introduction
Page: page 2
In addition to logic devices, there are analog, RF and input/output (I/O) devices on the same piece, all of which have channel lengths of more than 100nm [16,17].
Point 6: Figure 1 (a): HKMG => “High K metal gate (HKMG)” should be explained in the figure caption.
ICP and ALD also should be defined in the figure captions.
Response 6: We sincerely thank the reviewer for careful reading. As suggested by the reviewer, we have explained the HKMG, ICP and ALD in the Figure 1.
Corresponding change in manuscript: Yes
Location of Change:
Section: Part2. Materials and Methods
Page: page 3
Figure 1. (a) The main fabrication process flow of vertically stacked horizontal nanosheets. (b) 3D schematic diagram of the test pattern after the step of hard mask removal. (c) Summary of the parameters of the test pattern. (Inductively coupled plasma: ICP; High k dielectric and metal gate: HKMG; Atomic layer deposition: ALD)
Point 7: Sentaurus TCAD simulator => Sentaurus Technology Computer Aided Design (TCAD) simulator
Response 7: Thanks for your correction. According to your nice suggestions, we have made corresponding modification in Abstract of our previous manuscript.
Corresponding change in manuscript: Yes
Location of Change:
Section: Abstract
Page: page 1
The stress of the nanosheets in the stacking direction increased first and then decreased during the process of channel release by technology computer aided design (TCAD) simulation.
Point 8: 0.3GPade ??? => 0.3 GPa
Response 8: Thanks for your careful checks. We are sorry for our carelessness. Based on your comments, we have corrected the “0.3Gpade” into “0.3GPa”.
Corresponding change in manuscript: Yes
Location of Change:
Section: Part2. Materials and Methods
Page: page 3
Then, tensile stress is generated in the Si layers of the pad, and a compressive stress of about 0.3GPa is generated in the Si layers of the channel.
Point 9: Figure 3(a): "Middle and Bottom" end is partially overlapping with the black solid line. (Please adjust it)
Response 9: Thanks for your careful checks. We are sorry for our carelessness. Based on your comments, we have made corresponding adjustments to Figure 3(a).
Corresponding change in manuscript: Yes
Location of Change:
Section: Part3. Results and Discussion
|
Page: page 5
|
|
Figure 3. (a) The stress evolution on the 60nm-wide nanosheets along with increasing etching amount of the SiGe layers. Inset: The deformation of the nanosheets caused by the maximum stress values obtained by finite element simulation. TEM cross-sectional images of 60nm-wide stacked nanosheets obtained by (b) single-step etching and (c) multi-step etching.
Point 10: Line 199: Selectivity => Please define it in this spot by a simple equation
Response 10: Thanks for your suggestion. “Selectivity is defined as the ratio between the SiGe tunnel depth (d) and Si layer thickness loss ((a-b)/2) once the tunnel configuration is achieved, that is , which is illustrated in Figure 4.” We have added this sentence to our previous manuscript. Meanwhile, for the reader's convenience, we have embedded the selectivity equation in Figure 4.
Corresponding change in manuscript: Yes
Location of Change:
Section: Part3. Results and Discussion
Page: page 5-6
Selectivity of ~ 168:1 for our novel channel release process is determined using long etch times on Si0.7Ge0.3 for extremely wide nanosheet. Selectivity is defined as the ratio between the SiGe tunnel depth (d) and Si layer thickness loss ((a-b)/2) once the tunnel configuration is achieved, that is , which is illustrated in Figure4. As shown in Figure 4, the SiGe is laterally consumed of 41.93nm, whereas etching Si nanosheet sidewall consumed of 0.25nm.
Figure 4. SiGe versus Si etching selectivity of multi-step etching with application oxidation treatment.
Point 11: Line 214: the TEM => the Transmission Electron Microscope (TEM)
Response 11: Thanks for your correction. According to your nice suggestions, we have made corresponding modification in part2 of our previous manuscript.
Corresponding change in manuscript: Yes
Location of Change:
Section: Part2. Materials and Methods
Page: page 3
The cross-section of the channel was analyzed by a high-resolution transmission electron microscope (TEM) to check the mechanical stability of stacked nanosheets.

Reviewer 5 Report
This manuscript reports a novel channel release process to fabricate nanosheet FET. This process consists of multi-step etching instead of conventional single-step etching. Stacked nanosheet with equal space were successfully fabricated. And mechanism was explained by the TCAD simulation. I feel that the presented results are not sufficient to publish the manuscript in Nanomaterials. From my point of view, I encourage authors to submit this manuscript to another journal(Semiconductor Science and Technology etc.). Here are my main concerns:
(1) I think the primary novelty here is that the novel channel release process based on multi-step etching. However, in the manuscript, the analysis and data are not sufficient to claim this novelty. The novelty of this manuscript is weak.
(2) Overall the manuscript is badly written. It is very often difficult to understand what the author mean, or the given information is incomplete. Similar contents and sentences are repeated. In addition, Result and Discussion section is not clear. In particular, it is very difficult to understand what the strengths of the proposed process are.
(3) There is no calibration between simulated data and experimental result. It would be better to perform calibration using experimental result.
Author Response
Response to Reviewer 5 Comments
Thank you very much for your time and valuable suggestions. We have considered your comments very carefully and made the corresponding modifications in the revised manuscript.
Point 1: I think the primary novelty here is that the novel channel release process based on multi-step etching. However, in the manuscript, the analysis and data are not sufficient to claim this novelty. The novelty of this manuscript is weak.
Response 1: Thank you for your valuable suggestion. As described in the part1.Introduction. When transistor architecture changes from FinFET to GAA-FET, the main conduction surface orientation is changed from the crystallographic plane (110) to (100), which results in an increase in electron mobility but a decrease in hole mobility [8]. To achieve N/P current matching in GAA-FET, it is necessary to introduce compressive stress in the p-channel metal–oxide–semiconductor (PMOS) channel to enhance hole mobility [9-11]. Selective epitaxial growth of SiGe in source and drain is recognized as one of the most effective methods to apply uniaxial compressive stress into PMOS channel [12,13]. According to the mainstream GAA-FET fabrication process flow, the process of channel release is per-formed after the selective epitaxial growth of SiGe source and drain [1]. Therefore, the Si nanosheets are subjected to compressive stress from the source and drain during the process of channel release. It is reported that Young’s modulus of Si nanosheet is much smaller than that of bulk Si [13], which means that Si nanosheet is more easily deformed under stress. Logic devices with shorter channel length may be less affected by sustained compressive stress during the process of channel release. In addition to logic devices, there are analog, RF and input/output (I/O) devices on the same piece, all of which have channel lengths of more than 100nm [16,17].
The experimental results show that when the source/drain exerts compressive stress on the channel, the long channel device will have destructive deformation after the channel release process. It is generally accepted that when the compressive stressors are introduced into the source/drain, the device with larger channel length is more prone to deformation after channel release process. In this paper, a novel channel release process based on multi-step etching was proposed. The multi-step etching process can realize the gradual release of stress to avoid the deformation of the nanosheet in the long channel device. This has not been reported in the existing literature. In addition, the plasma-free oxidation treatment was introduced in the multi-step etching process to realize selectivity of 168:1 for Si0.7Ge0.3 versus Si. The proposed novel process could realize channel release of nanosheets with multi-width from 30nm to 80nm with little Si loss.
Based on the above discussion, it can be found that the novel channel release process proposed in this paper can realize the monolithic integration of nanosheets of various sizes (especially for devices with channel length greater than 100nm) with the introduction of compressive stressor in the source/drain, unlocking the full potential of gate-all-around technology for digital, analog, and radio-frequency (RF) circuit applications.
Point 2: Overall the manuscript is badly written. It is very often difficult to understand what the author mean, or the given information is incomplete. Similar contents and sentences are repeated. In addition, Result and Discussion section is not clear. In particular, it is very difficult to understand what the strengths of the proposed process are.
Response 2: Thank you for your valuable suggestion. As suggested by the reviewer, we have made appropriate changes to the English representation of our previous manuscript.
Point 3: There is no calibration between simulated data and experimental result. It would be better to perform calibration using experimental result.
Response 3: Thank you for your valuable suggestion. In our previous work [8], process simulations have been carefully calibrated to ensure the accuracy of stress simulations. Our previous work has been referenced in this paper, so the TCAD model calibration is not explained in this paper.
In this work, the stress value obtained from the TCAD model is substituted into the finite element simulation model to obtain the deformation of the nanosheet. As described in this paper, the simulation results are consistent with the experimental results, which shows the accuracy of the experimental method.
Fig.2-2 in reference [8] shows the comparison results of our channel stress simulation results of Si0.5Ge0.5-channel GAA-FET and the reported data by IMEC in ref. [r1].
[r1] G. Eneman et al., "Stress Simulations of Fins, Wires and Nanosheets," ECS Transactions,
vol. 98, no. 5, pp. 253-265, 2020.

Round 2
Reviewer 2 Report
After reviewing revised manuscript entitled with “A Novel Si Nanosheet Channel Release Process for Fabrication of Gate-All-Around Transistors and its Mechanism Investigation”. Minor points have to revise before publication as follows.
1. GAAFET in the main-text should be revised to GAA-FET.
2. The resolution of Figure 1 and Figure 5 is not very clear.
2. The brand and model name of the RPCVD apparatus are not shown in the 2. Materials and Methods.
3. The Si/SiGe position in the left diagram of figure 4 is not right with the right picture of figure 4. Please check if the Si/SiGe position of the left image is a horizontal mirror image of the right picture.
4. Some wordings are suggested to revise. line 205, with application of oxidation treatment; line 208, the serious Si loss; line 220, the Si layer thickness loss; line 221, Figure 4.
Author Response
Response to Reviewer 2 Comments
(Round 2)
Thank you very much for your time and valuable suggestions. We have considered your comments very carefully and made the corresponding modifications in the revised manuscript.
Point 1: GAAFET in the main-text should be revised to GAA-FET.
Response 1: We sincerely thank the reviewer for careful reading. As suggested by the reviewer, we have made extensive corrections to our previous manuscript.
Corresponding change in manuscript: Yes
Section: Part1. Introduction, Part3. Results and Discussion
Page: page 1, page 2, page 4
As the technology node scales down to 3nm nodes and beyond, gate-all-around field effect transistor (GAA-FET) made of vertically stacked horizontal nanosheets is considered the most promising candidate for FinFET replacement [1-4]. First of all, GAA-FET has superior electrostatic controllability and improved short-channel effects suppression [5]. In addition, the design flexibility of GAA-FET allows a continuous range of nanosheet widths to satisfy different requirements in a single chip with a small areal cost [6]. GAA-FET can be fabricated with minimal changes from FinFET integration [7]
When transistor architecture changes from FinFET to GAA-FET, the main conduction surface orientation is changed from the crystallographic plane (110) to (100), which results in an increase in electron mobility but a decrease in hole mobility [8]. To achieve N/P cur-rent matching in GAA-FET, it is necessary to introduce compressive stress in the p-channel metal–oxide–semiconductor (PMOS) channel to enhance hole mobility [9-11]. Selective epitaxial growth of SiGe in source and drain is recognized as one of the most effective methods to apply uniaxial compressive stress into PMOS channel [12,13]. According to the mainstream GAA-FET fabrication process flow, the process of channel release is per-formed after the selective epitaxial growth of SiGe source and drain [1].
To obtain GAA-FETs with high electrical performance, it is expected to obtain stacked nanosheets with uniform spacing between adjacent nanosheets.
Point 2: The resolution of Figure 1 and Figure 5 is not very clear.
Response 2: Thanks for your careful checks. We have replaced Figure 1 and Figure 5 with higher resolution images.
Corresponding change in manuscript: Yes
|
Page: page 1, page 6
|
|
Figure 1. (a) The main fabrication process flow of vertically stacked horizontal nanosheets. (b) 3D schematic diagram of the test pattern after the step of hard mask removal. (c) Summary of the parameters of the test pattern. (Inductively coupled plasma: ICP; High k dielectric and metal gate: HKMG; Atomic layer deposition: ALD)
Figure 5. Schematic diagram of the multi-step etching process for channel release of single width nanosheets with application of oxidation treatment.
Point 3: The brand and model name of the RPCVD apparatus are not shown in the 2. Materials and Methods.
Response 3: I am very sorry that we did not mark clearly in the manuscript. The Si/SiGe multilayer wafer used in this paper is a commercially available wafer, so we do not know the brand and model name of the RPCVD apparatus. Thanks to the valuable suggestions of the reviewers, we have made changes in the revised manuscript.
Corresponding change in manuscript: Yes
Section: Part2. Materials and Methods
Page: page 3
First, three cycles of Si/SiGe multilayer were deposited on an 8-inch Si substrate using a reduced pressure chemical vapor deposition (RPCVD) apparatus, where the thickness of the Si and SiGe layers were around 9 nm, and the Ge composition in the SiGe layers was 30% (The Si/SiGe multilayer wafer used in this paper is a commercially available wafer.). Then, a hard mask was grown on the Si/SiGe multilayer and the designed test pattern was transferred to the hard mask.
Point 4: The Si/SiGe position in the left diagram of figure 4 is not right with the right picture of figure 4. Please check if the Si/SiGe position of the left image is a horizontal mirror image of the right picture.
Response 4: We sincerely thank the reviewer for careful reading. As suggested by the reviewer, we have modified Figure 4 accordingly.
Corresponding change in manuscript: Yes
Section: Part3. Results and Discussion
Page: page 5
Figure 4. SiGe versus Si etching selectivity of multi-step etching with application of oxidation treatment.
Point 5: Some wordings are suggested to revise. line 205, with application of oxidation treatment; line 208, the serious Si loss; line 220, the Si layer thickness loss; line 221, Figure 4.
Response 5: We sincerely thank the reviewer for careful reading. As suggested by the reviewer, we have made extensive corrections to our previous manuscript.
Corresponding change in manuscript: Yes
Section: Part3. Results and Discussion
Page: page 5, page 6
Figure 4. SiGe versus Si etching selectivity of multi-step etching with application of oxidation treatment.
Based on the concept of multi-step etching, oxidation treatment was introduced to solve the problem of the serious Si loss caused by inevitable over-etching.
Selectivity is defined as the ratio between the SiGe tunnel depth (d) and the Si layer thickness loss ((a-b)/2) once the tunnel configuration is achieved, that is , which is illustrated in Figure 4.
